# Reversible neuropathic pain model created by long-term optogenetic nociceptor stimulation using light-responsive pain mice

Satoshi Kouroki[1☯], Toyoaki Maruta [1*☯], Kotaro Hidaka[1], Tomohiro Koshida[1], Mio Kurogi[1], Yohko Kage[2], Ayako Miura[2], Hikaru Nakagawa[2], Toshihiko Yanagita[3], Ryu Takeya [2], Isao Tsuneyoshi[1]

1 Department of Anesthesiology, Faculty of Medicine, University of Miyazaki, Miyazaki, Miyazaki, Japan,
2 Department of Pharmacology, Faculty of Medicine, University of Miyazaki, Miyazaki, Miyazaki, Japan,
3 Department of Clinical Pharmacology, Faculty of Medicine, School of Nursing, University of Miyazaki, Miyazaki, Miyazaki, Japan

☯ These authors contributed equally to this work.
* mmctm2@yahoo.co.jp

## Abstract

Neuropathic pain has a significant social impact, with high morbidity and reduced productivity, the underlying mechanisms of neuropathic pain remain poorly understood, and effective therapeutic strategies remain elusive. The development of animal models of neuropathic pain that stimulate only the nociceptors and not the other sensory receptors or motor nerves is desirable for elucidating the complex pathogenesis of neuropathic pain. We have previously reported the generation of $Na_V1.7$−channelrhodopsin-2 (ChR2), $Na_V1.8$−ChR2, and $Na_V1.9$−ChR2 mice. Optogenetics was employed in these light-responsive pain mice for generating nociceptive pain by specifically exciting the spinal dorsal root ganglion neurons, in which the respective $Na^+$ channels are expressed through exposure to blue light. This study aimed to compare the neuropathic pain produced by the prolonged exposure of light-responsive pain mice to blue light. A reversible neuropathic pain state was established persisting for a minimum of 24 hours when each light-responsive pain mouse was irradiated with light of an intensity that consistently elicited pain. Furthermore, the mice also showed pain sensitivity to light irradiation and mechanical stimulation. The expression of c-Fos, a marker for neuronal activity following noxious stimulation, was increased in the dorsal horn of the spinal cord on the light irradiated side. DS-1971a, a selective $Na_V1.7$ inhibitor, was effective in attenuating neuropathic pain in all light-responsive pain mice. In conclusion, optogenetics helps elucidate the specific functions of sodium channel subtypes in pain signaling, thereby advancing our understanding and paving the way for the development of further effective treatments for pain disorders in the future.

**Data availability statement:** All relevant data are within the manuscript and its Supporting Information files.

**Funding:** This research was supported by Japan Society for the Promotion of Science (JSPS): KAKENHI Grant Numbers 18K08859, 21K08925, and 16H06276 (Advanced Animal Model Support), and a Grant-in-Aid for Clinical Research from Miyazaki University Hospital. The funders had no role in study design, data collection and analysis, decision to publish, or preparation of the manuscript.

**Competing interests:** The authors have declared that no competing interests exist.

## Introduction

Neuropathic pain, which affects 7–8% of the population, is considered a social problem as it results in decreased productivity [1]. However, the pathogenesis of neuropathic pain has not been fully elucidated; thus, no treatment has been established for managing neuropathic pain [1,2]. Multiple factors, which are modifiable over time, contribute to the onset of neuropathic pain. Thus, unraveling the complex pathogenesis of neuropathic pain is challenging. The most widely employed models of neuropathic pain attributed to peripheral neuropathy in rodents are physical injury models of peripheral nerves, including the spared nerve injury (SNI), chronic constriction injury (CCI), partial sciatic nerve ligation (PSL), partial sciatic nerve ligation (PSL), and spinal nerve ligation (SNL) (Chung model) models [3]. Although these models produce allodynia, they can cause motor and/or non-pain sensory neuropathies. Therefore, despite the development of various animal models of neuropathic pain in the field of pain research, the creation of models by stimulating only the nociceptors and not the other sensory receptors or motor nerves is desirable.

The $Na_V1.7$, $Na_V1.8$, and $Na_V1.9$, which are voltage-gated $Na^+$ channel subtypes, are predominantly expressed in the spinal dorsal root ganglion (DRG) neurons and are involved in pain signaling and neuropathic pain development [4]. In the voltage-gated $Na^+$ channels, which consist of $Na_V1.1$-1.9 subtypes, each subtype possesses different kinetics and expression patterns, which are reflected in the functional groupings of the peripheral sensory neurons [5]. Large DRG neurons with myelinated Aβ and Aδ fibers express mainly tetrodotoxin-sensitive (TTX-S) $Na_V1.1$, $Na_V1.6$ and $Na_V1.7$. A subset of these neurons also express tetrodotoxin-resistant (TTX-R) $Na_V1.8$, which may correspond to Aβ nociceptors. In contrast, small diameter nociceptive neurons with unmyelinated C fibers express high levels of TTX-R $Na_V1.8$ and $Na_V1.9$, and TTX-S $Na_V1.7$ and $Na_V1.6$. These differences in the $Na^+$ channel expression are reflected in the differences in the morphology of the action potential waveform: $Na_V1.7$ contributes to the rising phase of the action potential and amplifies subthreshold stimulation, while $Na_V1.8$ contributes mainly to the rising phase and $Na_V1.9$ amplifies subthreshold stimuli. Furthermore, $Na_V1.7$, $Na_V1.8$, and $Na_V1.9$ have been reported to cause abnormal pain or painlessness in humans owing to gain-of-function or loss-of-function mutations [5].

Recently, optogenetics using light-responsive ion channels known as opsins has facilitated selective activation and inhibition of the target neurons *in vivo* [6]. This technique of optogenetics has become widely used in neuroscience and has proven useful in elucidating complex pain pathways in the field of pain research [7–12]. We have developed $Na_V1.7$−channelrhodopsin-2 (ChR2), $Na_V1.8$−ChR2, and $Na_V1.9$−ChR2 mice and applied optogenetics to specifically excite neurons that exhibit expression of each $Na^+$ channel [13,14]. When the plantar of these light-responsive pain mice were irradiated with blue light, nociceptive pain was produced in each mouse at varying light intensities [13]. Previously, Daou et al. reported the occurrence of transient neuropathic pain following long-term exposure to blue light in $Na_V1.8$−ChR2 mice [12]. Such optogenetics-based neuropathic pain models are superior

to conventional animal models as they are able to target only the sensory nerves responsible for pain. In this study, we compared the neuropathic pain produced by long-term and continuous exposure of $Na_V1.7$−ChR2, $Na_V1.8$−ChR2, and $Na_V1.9$−ChR2 mice to blue light.

## Materiavls and methods

### Animals

Wild-type (WT) C57BL/6J mice, commonly known as B6J mice, and Ai32 mice (C57BL/6 background) were purchased from the Jackson Laboratory (Bar Harbor, ME, USA). All the mice were individually housed in a temperature and humidity-controlled environment with a 12-h light-dark cycle, and were permitted free access to food and water. This study was conducted in strict accordance with the guidelines for the Proper Conduct of Animal Experiments (Science Council of Japan) and approved by the Experimental Animal Care and Use Committee of the University of Miyazaki (Permit Number: 2024-511). Male mice between 2 and 6 months old were used in the experiments. All efforts were made to minimize the number of animals used and their suffering. Because all mice recover to a normal, pain-free state, euthanasia need not be considered. Mice in each group were randomly selected, and the experimenter was blinded to the mouse genotype and drug treatment.

### Production of genetically modified mice

$Na_V1.x$−ChR2 mice were created as described previously [13–15]. We produced the bicistronic founder generation (F0) of $Na_V1.7$–iCre ($Na_V1.7^{iCre/+}$), $Na_V1.8$–iCre ($Na_V1.8^{iCre/+}$), and $Na_V1.9$–iCre ($Na_V1.9^{iCre/+}$) knock-in mice using the CRISPR/Cas9 system, with financial and technical assistance from Advanced Animal Model Support (AdAMS). For each subtype, we targeted the 5′-gaaagcaggaaatagagctt-3′, 5′-cctggacctcagtgaagacactc-3′, or 5′-cttggatgtgcccaagatca-3′ sequence corresponding to the *Scn9a*, *Scn10a*, or *Scn11a* gene, respectively. Following DNA confirmation of the desired genetic modification, $Na_V1.7$–ChR2 ($Na_V1.7^{iCre/+}$;Ai32/+), $Na_V1.8$–ChR2 ($Na_V1.8^{iCre/+}$;Ai32/+), and $Na_V1.9$–ChR2 ($Na_V1.9^{iCre/+}$;Ai32/+) mice were generated by breeding homozygous $Na_V1.7$–iCre ($Na_V1.7^{iCre/iCre}$), $Na_V1.8$–iCre ($Na_V1.8^{iCre/iCre}$), or $Na_V1.9$–iCre ($Na_V1.9^{iCre/iCre}$) mice with homozygous Ai32 mice, carrying the *ChR2(H134R)-EYFP* gene in the *Gt(ROSA)26Sor* locus. The *ChR2(H134R)-EYFP* gene was capable of expression through its CAG promoter by eliminating the loxP-flanked transcriptional STOP cassette after breeding with iCre mice.

### von Frey test for determining mechanical sensitivity

Mechanical sensitivity was examined by determining the paw withdrawal threshold using an electronic von Frey esthesiometer (IITC Life Science Inc., Woodland Hills, CA, USA) fitted with a polypropylene tip. Each adult mouse was placed in a 10 cm × 10 cm suspended chamber with a metallic mesh floor. After acclimation of the mice for 30 min, a polypropylene tip was applied perpendicularly to the plantar surface of the right and left hind paws with sufficient force for 3–4 s. Brisk withdrawal or paw flinching was considered a positive response. The pain threshold was calculated as the mean of three measurements.

In this study, the left and right hindpaws of $Na_V1.x$–ChR2 mice were tested for sensitivity to mechanical stimuli before and after (1, 3, 6, 24, 36, and 48 h) a prolonged (30 min), suprathreshold (5 and 7.5 mW for $Na_V1.7$–ChR2, 1.2 and 1.8 mW for $Na_V1.8$–ChR2, and 1.5 and 2.25 mW for $Na_V1.9$–ChR2) blue-light stimulation of the ipsilateral hindpaw. The blue-light intensity employed in this experiment was the intensity at which the paw withdrawal frequency was nearly 100% in the previously reported blue-light plantar irradiation test and 1.5 times that intensity [13]. The light intensity was determined using a light power meter (LPM-100™; Bioresearch Center Inc., Aichi, Japan). The non-stimulated contralateral hindpaw was employed as an internal control. Mice were anesthetized with 3.0% sevoflurane during the 30-min stimulation. Laser was pulsed at 2 Hz with 100 ms pulse duration. The settings for these continuous irradiations were determined by a preliminary reported study [12].

To determine the analgesic effects of DS-1971a, a selective $Na_V1.7$ inhibitor, on the long-term optogenetic stimulation-induced neuropathic pain, von Frey test was performed. One side of the hindpaw of $Na_V1.x$ –ChR2 mice was tested for sensitivity to mechanical stimuli before and 1 hour after a prolonged (30 min), suprathreshold (7.5 mW for $Na_V1.7$–ChR2, 1.8 mW for $Na_V1.8$–ChR2, and 2.25 mW for $Na_V1.9$–ChR2) blue-light stimulation of the ipsilateral hindpaw. DS-1971a (10 mg/kg and 100 mg/kg, Daiichi Sankyo, Tokyo, Japan) in 0.5% methylcellulose or vehicle (0.5% methylcellulose) was subsequently administered orally. The pain threshold in the ipsilateral hindpaw was measured 2 hours after DS-1971a administration. The dose levels of DS-1971a administration were determined by a preliminary reported study [16], in which DS-1971a at 10 mg/kg and 100 mg/kg successfully reduced pain thresholds over a long period of time in the spinal nerve ligation and partial sciatic nerve ligation mouse models.

### Light irradiation test to determine the light-responsive hypersensitivity due to long-term optogenetic stimulation

Light irradiation test was performed before and 1 hour after a prolonged (30 min), suprathreshold (7.5 mW for $Na_V1.7$–ChR2, 1.8 mW for $Na_V1.8$–ChR2, and 2.25 mW for $Na_V1.9$–ChR2) light stimulation. Mice were habituated for 1 h in transparent cubicles (10 cm × 6.5 cm × 6.5 cm) set atop a 5-mm-thick glass floor and separated from each other by opaque dividers. Acute nocifensive behaviors were elicited by light from a pulsing light-emitting diode (LED) (465 nm blue light at 10 Hz; Doric Lenses Inc., Quebec, Canada) set at different intensities and aimed at the plantar surface of the hind paw. The light intensity was determined using a light power meter (LPM-100™). Since the power meter measures light intensity in milliwatts (mW), the light density was calculated in $mW/mm^2$ by dividing the light intensity by the illuminated area in square millimeters ($48 mm^2$). The mice underwent five trials of 1 s each, with 5-s intervals between the trials. The percentage of trials demonstrating hind paw withdrawal or paw licking was recorded.

### Immunohistochemistry

Mice were confirmed hypersensitive by von Frey test 1 hour after a prolonged (30 min), suprathreshold (7.5 mW for Nav1.7–ChR2, 1.8 mW for $Na_V1.8$–ChR2, and 2.25 mW for $Na_V1.9$–ChR2) light stimulation. The mice were subsequently anesthetized with sevoflurane and intracardially perfused with 50 mL perfusion buffer, followed by 100 mL 4% paraformaldehyde (PFA) in phosphate-buffered saline (PBS) (pH 7.4) at room temperature for 30 min. The L3-5 lumbar spinal cord was dissected and post-fixed in 4% PFA for 2 h at 4 °C, cryoprotected in 30% sucrose in PBS, and incubated overnight at 4 °C; 16-µm thick sections were cut from the freeze-fixed spinal cord with the temperature maintained at –20 °C using a cryostat (Leica Biosystems, Nussloch, Germany). Samples of spinal cord were placed directly on slides. The spinal cord sections were incubated in 0.1% Triton X-100 and 5% goat serum in PBS at room temperature for 4 h, followed by incubation with rabbit anti-c-Fos recombinant monoclonal antibody (1:500; catalog #ab222699, Abcam, Cambridge, MA, UK) at 4 °C with overnight agitation. Thereafter, the sections were washed thrice with PBS, followed by incubation with goat anti-rabbit IgG (H+L) (Alexa Fluor™ Plus 594, 1:300; catalog #A32740, Invitrogen, Waltham, MA, USA) for 1 h at room temperature. These sections were washed, air-dried, and mounted with a coverslip using an antifade mounting medium (Mowiol™ 4-88, catalog #81381, Sigma-Aldrich, St Louis, MO, USA). The fluorescence of the transgenic ChR2–EYFP was sufficient for visualization without immunostaining. The prepared slides were stored at 4 °C until further examination. The morphology of the different tissues was analyzed using a BZ-9000 fluorescence microscope (Keyence, Osaka, Japan). Images were processed using ImageJ software (NIH, Bethesda, MD, USA) to optimize brightness and contrast [17].

### Statistical analysis

Each behavioral experiment evaluated $n \geq 10$ animals, whereas the examinations of c-Fos expression evaluated $n = 6$ animals. For the behavioral experiments, data were analyzed using paired $t$-test or one-way analysis of variance (ANOVA) followed by Bonferroni post-hoc analysis. The results are presented as mean ± standard deviation (SD). Statistical

significance was set at $P < 0.05$. The statistical software JMP Pro 17 (SAS Institute, Inc., Cary, NC, USA) for Macintosh was used for the statistical analyses.

## Results

### Mechanical sensitization induced by long-term optogenetic stimulation

To examine whether long-term optogenetic stimulation induces mechanical sensitization, prolonged suprathreshold blue-light irradiation (30 min, 2 Hz, two types of intensity) to the hindpaw plantar surface of anesthetized $Na_V1.x$–ChR2 mice was performed. As shown in Fig 1a,b, and c, compared with the contralateral hindpaw, significant mechanical hypersensitivity was observed in the ipsilateral hindpaw of $Na_V1.x$–ChR2 mice. This mechanical hypersensitivity was reversible and persisted for longer periods at higher light intensities.

### Light-responsive hypersensitivity due to long-term optogenetic stimulation

To confirm peripheral sensitization by prolonged optogenetic stimulation, we examined the blue LED light irradiation hind paw withdrawal test before and 1 hour after prolonged (30 min) optogenetic stimulation. The leftward shift of the light power-withdrawal curve before and after prolonged optogenetic stimulation indicated that prolonged optogenetic stimulation hypersensitized the response to nociceptive pain induced by light irradiation in the $Na_V1.x$–ChR2 mice (Fig 2).

### c-Fos expression in the spinal cord dorsal horn following long-term optogenetic stimulation

To investigate whether light irradiation of the hindpaw plantar is transmitted from the peripheral nerve to the central nervous system as a nociceptive signal, we investigated c-Fos expression in the secondary sensory neurons in the dorsal horn of the spinal cord. Mice that demonstrated hypersensitivity using the von Frey test were fixed for

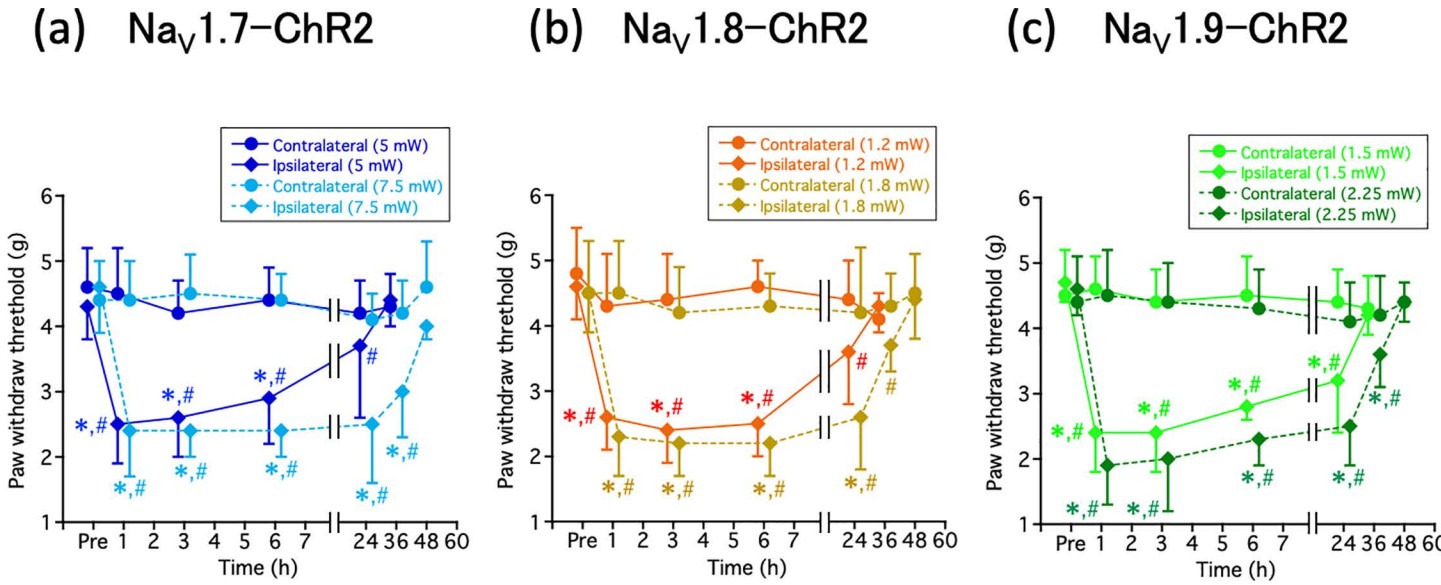

**Fig 1. Paw withdrawal test (von Frey test) in $Na_V1.7$–ChR2 (a), $Na_V1.8$–ChR2 (b), and $Na_V1.9$–ChR2 mice.** The von Frey test was performed with (a) NaV1.7–ChR2, (b) NaV1.8–ChR2, and (c) NaV1.9–ChR2 mice before (Pre) and after (1, 3, 6, 24, 36, and 48 h) light stimulation. Prolonged, suprathreshold blue-light exposure (30 min, 2 Hz, two types of intensity) to the ipsilateral hindpaw of anesthetized $Na_V1.x$ –ChR2 mice produces long-term mechanical hypersensitivity lasting up to 24 h and 36 h after stimulation, respectively. The hind paw withdrawal data were analyzed using paired $t$-test (ipsilateral compared with contralateral) or one-way ANOVA followed by Bonferroni post-hoc analysis (each time compared with Pre). All the results are calculated as mean ± SD of 10 or more animals. *$P < 0.05$, compared with contralateral. #$P < 0.05$, compared with before optogenetic stimulation (Pre).

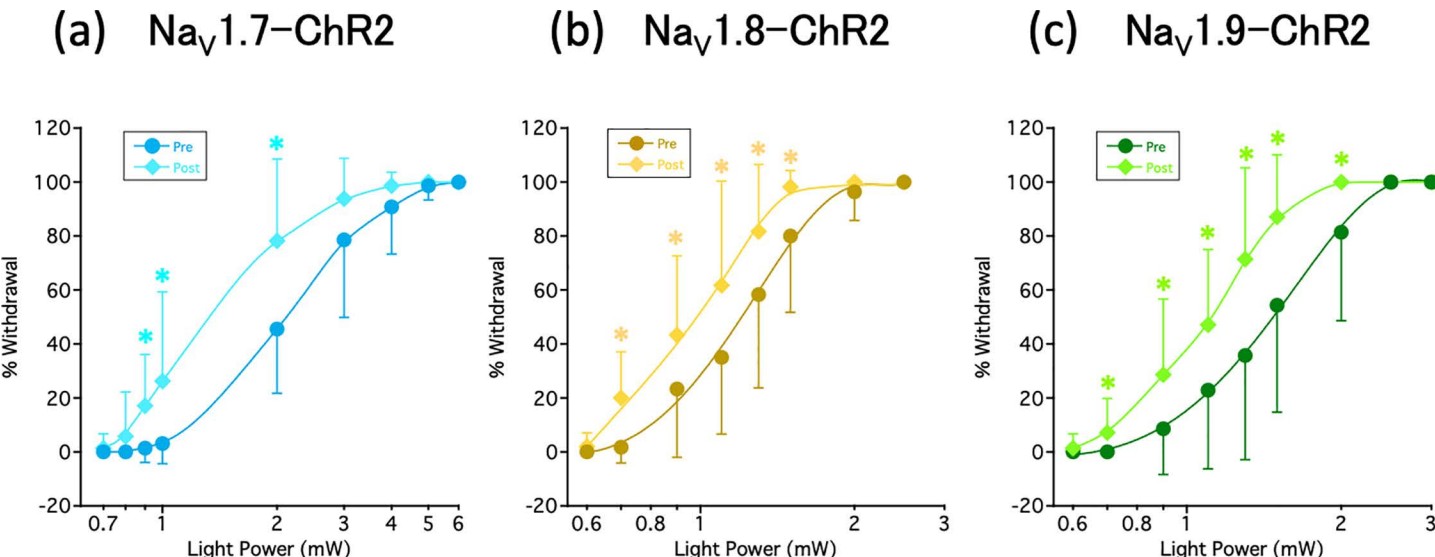

**Fig 2. Light irradiation hind paw withdrawal test before and after long-term optogenetic stimulation.** The blue light irradiation hind paw withdrawal test was performed before (Pre) and 1 hour after (Post) a prolonged (30 min), suprathreshold (7.5 mW for $Na_V1.7$–ChR2, 1.8 mW for $Na_V1.8$–ChR2, and 2.25 mW for $Na_V1.9$–ChR2) light stimulation in (a) $Na_V1.7$–ChR2, (b) $Na_V1.8$–ChR2, (c) $Na_V1.9$–ChR2 mice. The data were analyzed using paired *t*-test. All the results are calculated as mean ± SD of 10 or more animals. *$P < 0.05$, compared with Pre.

immunofluorescence analysis. We observed c-Fos expression in the ipsilateral and contralateral dorsal horn of the lumber spinal cord after long term optogenetic stimulation (Fig 3). In the spinal cord dorsal horn of $Na_V1.x$–ChR2, compared to the contralateral side, c-Fos expression was increased in the ipsilateral side.

### Analgesic effect of DS-1971a on long-term optogenetic stimulation-induced neuropathic pain

Clinical trials of selective $Na^+$ channel inhibitors are underway for the treatment of acute postoperative pain and neuropathic pain [18]. We have examined the analgesic effect of 10 and 100 mg/kg DS-1971a, which is a selective $Na_V1.7$ inhibitor [16], on long-term optogenetic stimulation-induced neuropathic pain in $Na_V1.x$–ChR2 mice. DS-1971a at both 10 mg/kg and 100 mg/kg almost completely suppressed long-term optogenetic stimulation-induced neuropathic pain in the $Na_V1.7$-ChR2, $Na_V1.8$-ChR2 and $Na_V1.9$-ChR2 mice (Fig 4).

### Discussion

In this study, we demonstrated that a reversible neuropathic pain state was established persisting for a minimum of 24 hours when light-responsive pain mice ($Na_V1.x$-ChR2 mice) was irradiated with light of an intensity that consistently elicited pain. Furthermore, the mice also showed hypersensitivity to light irradiation and mechanical stimulation. The c-Fos expression increased in the dorsal horn of the spinal cord on the light irradiated side. DS-1971a, a selective $Na_V1.7$ inhibitor, was effective in attenuating neuropathic pain in all light-responsive pain mice.

In our previous studies [13,14], we generated bicistronic $Na_V1.7$−, $Na_V1.8$−, and $Na_V1.9$−iCre recombinase-expressing iCre recombinase under the endogenous $Na_V1.x$ gene promoter without disruption of $Na_V1.7$, $Na_V1.8$, and $Na_V1.9$ by CRISPR/Cas9-mediated homologous recombination. Furthermore, by crossing these lines with homozygous Ai32 mice via the Cre-LoxP system, transgenic $Na_V1.x$−ChR2 mouse lines ($Na_V1.x^{iCre/+}$;Ai32/+) were generated in which ChR2 was expressed only in $Na_V1.x$-expressing sensory neurons. The mice expressing ChR2 in the $Na_V1.7$, $Na_V1.8$, and $Na_V1.9$ channels demonstrated a nociceptive response to blue light. Differences in this light sensitivity

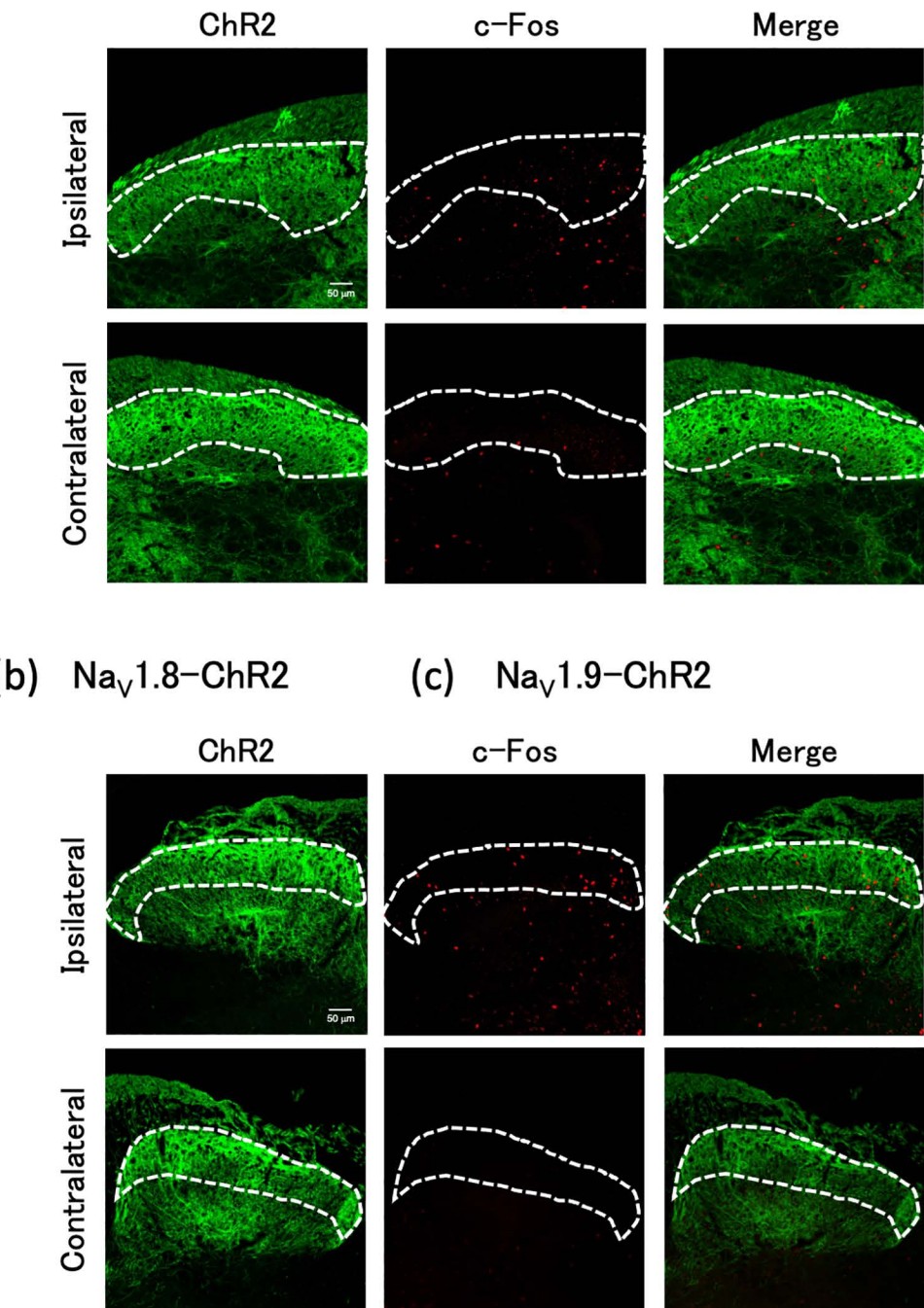

**Fig 3. c-Fos expression in the lumber spinal cord dorsal horn.** c-Fos expression in the ipsilateral and contralateral dorsal horn of the lumber spinal cord after long term (30 min) optogenetic stimulation (7.5 mW for Na$_V$1.7–ChR2, 1.8 mW for Na$_V$1.8–ChR2, and 2.25 mW for Na$_V$1.9–ChR2) are shown in (a) Na$_V$1.7–ChR2, (b) Na$_V$1.8–ChR2, (c) Na$_V$1.9–ChR2 mice. Scale bar, 50 μm. Quantification of c-Fos-positive neurons in laminae I–III of the L3–5 lumber spinal segments from optogenetically stimulated Na$_V$1.x–ChR2 mice are shown in (d). The data were analyzed using paired *t*-test. All results are calculated as mean ± SD of 6 animals. \*$P < 0.01$, compared with the contralateral.

## (c)  Na$_V$1.9–ChR2

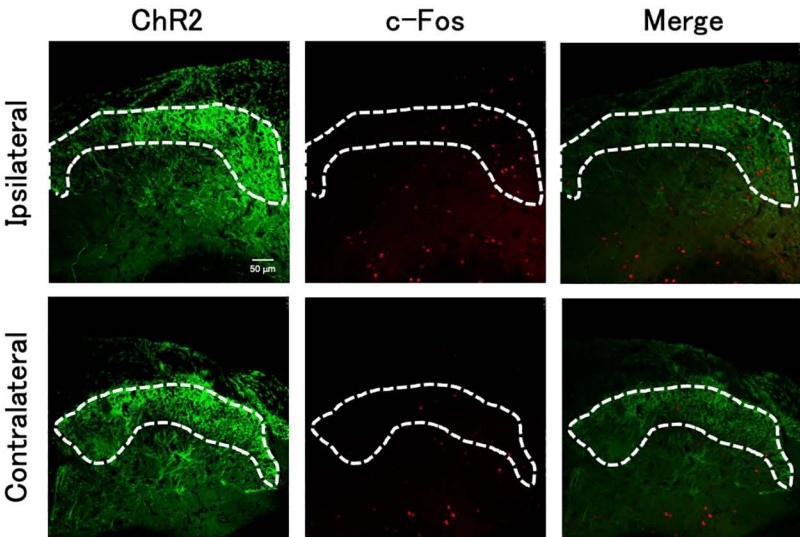

## (d)  c-Fos-positive neuron in laminae I–III

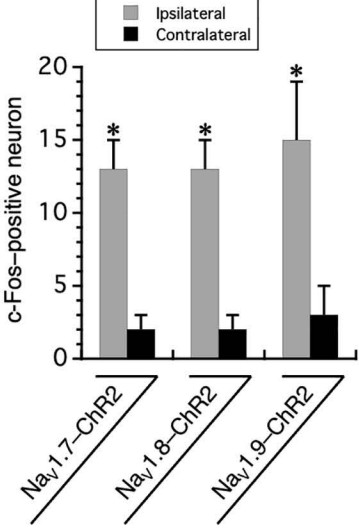

**Fig 3.** Continued.

were observed between Na$_V$1.x–ChR2 mice, indicating that this may depend on the expression and distribution of ChR2 in the DRG, or that light-sensitivity may reflect inherent disparities associated with the varying roles of each Na$^+$ channel subtype in pain transmission [13,14]. Daou et al. reported that prolonged light exposure in Na$_V$1.8–ChR2 mice resulted in an increase in c-Fos expression in the dorsal horn of the spinal cord, which in turn resulted in persistent

**(a)** Na$_V$1.7–ChR2

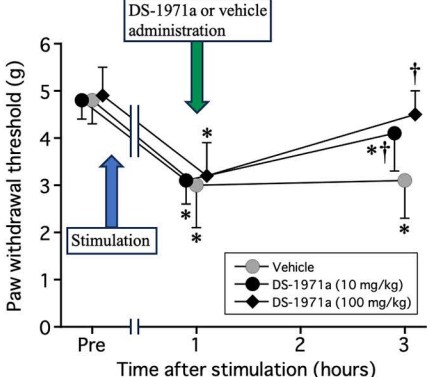

**(b)** Na$_V$1.8–ChR2

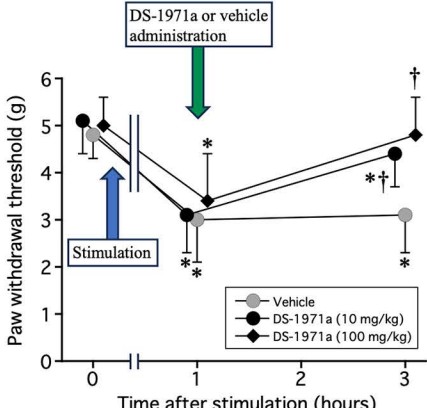

**(c)** Na$_V$1.9–ChR2

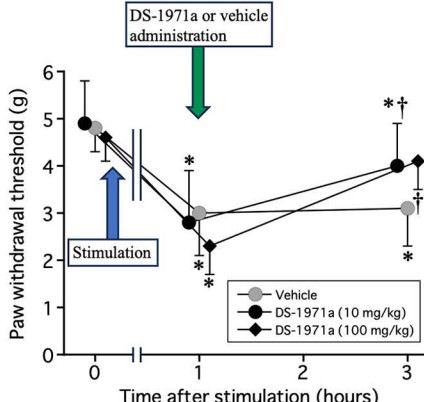

**Fig 4. Analgesic effect of DS-1971a on long-term optogenetic stimulation-induced neuropathic pain.** von Frey tests were performed in (a) NaV1.7–ChR2, (b) NaV1.8–ChR2, (c) NaV1.9–ChR2 mice before optogenetic stimulation (7.5 mW for NaV1.7–ChR2, 1.8 mW for NaV1.8–ChR2, and 2.25 mW for NaV1.9–ChR2), 1 hour after stimulation, and 3 hours after stimulation (2 hours after DS-1971a or vehicle administration). The data were analyzed using one-way ANOVA followed by Bonferroni post-hoc analysis. All data are calculated as mean ± SD of 10 animals. *$P < 0.05$, compared with pre-stimulation (time 0). †$P < 0.01$, compared with post-stimulation (1 hour after stimulation).

hypersensitivity to mechanical and thermal stimuli [12]. Similarly, as previously reported by Daou et al., we employed our light-responsive pain mice to demonstrate that prolonged light irradiation induces reversible pain hypersensitivity to mechanical stimuli in $Na_V1.x-ChR2$ mice (Fig 1). This is considered as a "windup phenomenon"; moreover, the fact that not only mechanical stimuli but also light irradiation caused pain hypersensitivity (Fig 2), and increased expression of c-Fos, a marker for neuronal activity following noxious stimulation, in the dorsal horn of the spinal cord on the light-irradiated side (Fig 3) collectively suggest that continuous light irradiation may repeatedly input nociceptive signals from peripheral nerves to the central nervous system, thereby leading to central sensitization [19]. In this study, the selective $Na_V1.7$ inhibitor DS-1971a for long-term optogenetic stimulation-induced neuropathic pain completely suppressed pain in the $Na_V1.x-ChR2$ mice. The fact that selective $Na_V1.7$ inhibitors are effective not only in $Na_V1.7-ChR2$ mice but also in $Na_V1.8-ChR2$ and $Na_V1.9-ChR2$ mice may be attributed to the global expression of $Na_V1.7$. Therefore, we demonstrated that selective $Na_V1.7$ inhibitors are effective in the management of pure neuropathic pain resulting from the stimulation of the nerves responsible for pain. In clinical practice, the use of $Na^+$ channel inhibitors that selectively act on peripheral nerves in the treatment of neuropathic pain may reduce adverse effects, such as arrhythmias and local anesthetic toxicity.

In the present study, we only observed the short-term analgesic effects of DS-1971a; however, successful initial treatment of neuropathic pain may lead to long-term analgesic effects. Furthermore, to understand the role of $Na_V1.8$ in neuropathic pain, it is important to clarify how selective $Na_V1.8$ inhibitors (although no selective $Na_V1.9$ inhibitors currently exist) act in the respective $Na_V1.x-ChR2$ mice. These limitations of the present study and issues should be addressed in future studies.

A multitude of animal models of neuropathic pain have been documented, and a substantial proportion of the knowledge pertaining to the biological alterations associated with neuropathic pain has been derived from these animal models [3,20,21]. Nevertheless, it is challenging to fully extrapolate findings from animal models to humans, as these models may not accurately recapitulate the clinical manifestations of the disease [20]. Several animal models of neuropathic pain have been developed through surgical damage to the sciatic nerve or adjacent nerves [3,21]. For example, the Chung model is typically created by ligating the fifth and sixth spinal nerves of the fourth through sixth lumbar spinal nerves that form the sciatic nerve. However, it is not possible to damage only nerve fibers expressing a specific voltage-gated $Na^+$ channel subtype with this method of creating animal models of neuropathic pain. The light-responsive pain mice can be employed to generate noninvasively neuropathic pain models through the specific stimulation of nerve fibers expressing particular voltage-gated $Na^+$ channel subtypes via prolonged light irradiation. Increased or hyperfunctional $Na^+$ channels not only enhance pain electrical signals but also alter intracellular signaling molecules, which are implicated in pain plasticity [22]. The use of light-responsive pain mice helps elucidate the exact intracellular pathogenesis of neuropathic pain, which involves an increase or hyperfunction of specific voltage-gated $Na^+$ channels. Furthermore, although pain persisted for up to 24–36 hours in this study, more chronic neuropathic pain may occur with higher light intensity or longer exposure duration, which may contribute to the understanding of chronic pain.

In conclusion, our study demonstrated that it is possible to create a reversible neuropathic pain model using light-responsive pain mice, $Na_V1.7-$, $Na_V1.8-$, and $Na_V1.9-ChR2$ mice, by prolonged light irradiation and repeated excitation of peripheral nociceptors. Optogenetics may further contribute to the elucidation of the specific function of sodium channel subtypes in pain signaling and facilitate the acquisition of more in-depth knowledge and the establishment of effective treatments for pain disorders in the future.

## Supporting information

**S1 File. Actual values in the respective graphs.** Actual values (mean±SD) in Figs 1,2,3d, and 4 were shown. (XLSX)

## Acknowledgments

This study was conducted in the Department of Anesthesiology, Faculty of Medicine, University of Miyazaki. The authors would like to extend their gratitude to Seiya Mizuno and Satoru Takahashi (Laboratory Animal Resource Center in Transborder Medical Research Center, Faculty of Medicine, University of Tsukuba, Tsukuba, Ibaraki, Japan) for the generation of genetically modified mice; Daiichi Sankyo Co., Ltd. for gifting DS-1971a; Noriko Hidaka and Kaori Kaji for their technical and secretarial assistance in this study; and Editage (www.editage.jp) for English language editing.

## Author contributions

**Conceptualization:** Toyoaki Maruta.

**Data curation:** Satoshi Kouroki, Toyoaki Maruta.

**Formal analysis:** Satoshi Kouroki, Toyoaki Maruta.

**Funding acquisition:** Satoshi Kouroki, Toyoaki Maruta.

**Investigation:** Satoshi Kouroki, Toyoaki Maruta, Kotaro Hidaka, Tomohiro Koshida, Mio Kurogi.

**Methodology:** Toyoaki Maruta, Yohko Kage, Ayako Miura, Hikaru Nakagawa.

**Project administration:** Toyoaki Maruta.

**Resources:** Satoshi Kouroki, Toyoaki Maruta.

**Writing – original draft:** Toyoaki Maruta.

**Writing – review & editing:** Toshihiko Yanagita, Ryu Takeya, Isao Tsuneyoshi.

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
