## [Decision Letter · Decision Letter 0]

18 Feb 2025

PONE-D-24-58742Reversible neuropathic pain model created by long-term optogenetic nociceptor stimulation using light-responsive pain micePLOS ONE

Dear Dr. Maruta,

Thank you for submitting your manuscript to PLOS ONE. After careful consideration, we feel that it has merit but does not fully meet PLOS ONE’s publication criteria as it currently stands. Therefore, we invite you to submit a revised version of the manuscript that addresses the points raised during the review process.

As you will see one of the reviewer recommends conducting additional experiments to strengthen the conclusions. If all of these experiments cannot be conducted, you must provide a detailed and comprehensive justification for their omission.

We look forward to receiving your revised manuscript.

Kind regards,

Julian Cheron

Academic Editor

PLOS ONE

2. Thank you for stating the following in the Acknowledgments Section of your manuscript: [This study was conducted in the Department of Anesthesiology, Faculty of Medicine, University of Miyazaki. This research was supported by JSPS KAKENHI Grant Numbers JP18K08859, 21K08925, and 16H06276 (AdAMS), and a Grant-in-Aid for Clinical Research from Miyazaki University Hospital. The authors would like to extend their gratitude to Seiya Mizuno and Satoru Takahashi (Laboratory Animal Resource Center in Transborder Medical Research Center, Faculty of Medicine, University of Tsukuba, Tsukuba, Ibaraki, Japan) for the generation of genetically modified mice; Noriko Hidaka and Kaori Kaji for their technical and secretarial assistance in this study; and Editage (www.editage.jp) for English language editing.]

Please remove any funding-related text from the manuscript and let us know how you would like to update your Funding Statement. Currently, your Funding Statement reads as follows: "This research was supported by Japan Society for the Promotion of Science (JSPS): KAKENHI Grant Numbers 18K08859, 21K08925, and 16H06276 (Advanced Animal Model Support), and a Grant-in-Aid for Clinical Research from Miyazaki University Hospital. The funders had no role in study design, data collection and analysis, decision to publish, or preparation of the manuscript."

Reviewers' comments:

Reviewer's Responses to Questions

**Comments to the Author**

1. Is the manuscript technically sound, and do the data support the conclusions?

Reviewer #1: Partly

Reviewer #2: Yes

2. Has the statistical analysis been performed appropriately and rigorously? 

Reviewer #1: Yes

Reviewer #2: Yes

3. Have the authors made all data underlying the findings in their manuscript fully available?

Reviewer #1: Yes

Reviewer #2: Yes

4. Is the manuscript presented in an intelligible fashion and written in standard English?

Reviewer #1: Yes

Reviewer #2: No

5. Review Comments to the Author

Reviewer #1: Experiments are well executed.

There is one important missing information whether they did a double blind approach for pharmacological treatments in the transgenic mice.

A more precise description of the clinical relevance of their model and additional experiments are needed to establish better comparisons with existing animal models and fully support their conclusions.

Reviewer #2: In the current manuscript the authors use optogenetic models to understand whether activation of distinct Na+ channels changes pain response upon further mechanical stimulation. In summary, the authors showed that opto stimulation generated a prolonged yet reversible pain response.

The experiments in the manuscript are simple and straightforward, convincingly showing that opto stimulation of the three types of Na+ channels does elicit changes in pain response.

The use of an inhibitor further proves the claim that this changes are indeed due to the optogenetic stimulation in this animal model.

Increased c-fos labeling upon opto genetic stimulation shows further proof that the model is indeed working and operating changes at a central level, and yields an interesting observation where neurons in the dorsal horn respond in a manner dependent on their ipsilateral/contralateral location.

Overall, this manuscript is well thought out and does give enough proof to sustain the claims.

Personally, I would just edit some parts of the text, both in sentence construction as well as in some details that seem to be less precise. As an example, the authors claim cfos to be a "a marker of pain perception" and although it might be true in this particular neuronal population/context, cfos is generally used as a proxy for neuronal firing/activation rather than a marker for pain perception.

6. PLOS authors have the option to publish the peer review history of their article (what does this mean? ). If published, this will include your full peer review and any attached files.

**Do you want your identity to be public for this peer review?** For information about this choice, including consent withdrawal, please see our Privacy Policy .

Reviewer #1: No

Reviewer #2: **Yes: ** Rui de Oliveira Beleza

---

## [Author Response · Author response to Decision Letter 1]

1 Mar 2025

Dear Dr. Julian Cheron,

Thank you for giving me the opportunity to submit a revised draft of the manuscript titled “Reversible neuropathic pain model created by long-term optogenetic nociceptor stimulation using light-responsive pain mice.” to PLOS ONE (Manuscript ID: PONE-D-24-58742).

We greatly appreciate the time and effort that you and the reviewers have dedicated to providing your valuable feedback on our manuscript. We are grateful to the reviewers for their insightful comments on this paper.

We have extensively revised our paper to reflect most of the suggestions provided by the reviewers. We have highlighted the changes within the manuscript in red font on marked version (Revised manuscript with Track Changes). Here is a point-by-point response to the reviewers’ comments and concerns.

Reviewer: 1

Major comments:

1) The use of “neuropathic pain” in the description of their model is confusing. Indeed, neuropathy is large family of pain types which can be acute, a chronic state resulting from neuronal lesion with sensory alterations, combined to inflammation or metabolic alterations... In the discussion, they compare the advantage of their acute model of nociception over sciatic nerve injury that leads to chronic neuropathy, a very different clinical picture. A more precise description of the proposed model in respect to clinical relevance therefore needs to be clarified to draw the right conclusions.

In this line, the authors should give examples and references about the other animal models of neuropathic pain they are comparing theirs to (introduction, line 53) to clarify latter the potential added value and limitations of their model. Also, they refer to a previous work of theirs introducing this optogenetic-based model (introduction, line 61-65) and mention that “nociceptive pain was produced”, not neuropathic pain….

Response: Thank you for your valuable comment. We have already provided examples and references about the other animal models of neuropathic pain, which we are comparing our model to, to clarify latter the potential added value of our model in the discussion section. Kindly see Discussion, Pages 27–29: lines 311–327, “A multitude of animal models of neuropathic pain…specific voltage-gated Na+ channels”.

Our light-responsive pain mice (NaV1.x−ChR2) induce nociceptive pain on exposure to specific wavelengths, similar to the light-responsive pain mice (NaV1.8−ChR2) developed by Daou et al. Moreover, repeated nociceptive pain can lead to neuropathic pain due to central sensitization [16]. Daou et al. and we have also demonstrated that prolonged light exposure can cause neuropathic pain. We have already mentioned these points in the discussion section (Page 26: lines 294-301, “Daou et al. reported that…thereby leading to central sensitization [16]”).

2) To comprehend the importance of each sodium channel they investigate, it is also important to describe better in the introduction their expression pattern in the nociceptive and sensory fiber types, their difference in terms of electrophysiology (for instance their different sensitivity to TTX), and the state of the art knowledge about their implication in neuropathic pain as their involvement is not utterly understood. Their role also needs to be described in line with the type of pain elicited by this optogenetic-induced nociceptor stimulation.

Response: Thank you for your valuable comment. As you have accurately pointed out, in order for the reader to understand the importance of the Na+ channels we are studying, the expression pattern of each Na+ channel, their electrophysiological differences and the latest findings on their involvement in neuropathic pain should be explained. However, these findings are lengthy and would better be summarized in a review article, and cannot be explained in the introduction; we have already mentioned the role of Na+ channels, citing the review by Wada et al. [3] (Pages 5–6: lines 55–57), and further we have added text on Na channels (Pages 6–7: lines 57–70) with an additional reference [4]. Readers of this paper will probably also read the preceding paper [12]. As the role of the Na channel is also described in that preceding paper, we feel that the description of the Na+ channels in this paper is sufficient.

Changes in the text:

“In the voltage-gated Na+ channels, which consist of NaV1.1-1.9 subtypes, each subtype possesses different kinetics and expression patterns, which are reflected in the functional groupings of the peripheral sensory neurons [4]. Large DRG neurons with myelinated Aβ and Aδ fibers express mainly tetrodotoxin-sensitive (TTX-S) NaV1.1, NaV1.6, and NaV1.7. A subset of these neurons also express tetrodotoxin-resistant (TTX-R) NaV1.8, which may correspond to Aβ nociceptors. In contrast, small diameter nociceptive neurons with unmyelinated C fibers express high levels of TTX-R NaV1.8 and NaV1.9, and TTX-S NaV1.7 and NaV1.6. These differences in the Na+ channel expression are reflected in the differences in the morphology of the action potential waveform: NaV1.7 contributes to the rising phase of the action potential and amplifies subthreshold stimulation, while NaV1.8 contributes mainly to the rising phase and NaV1.9 amplifies subthreshold stimuli. Furthermore, NaV1.7, NaV1.8, and NaV1.9 have been reported to cause abnormal pain or painlessness in humans owing to gain-of-function or loss-of-function mutations [4].” (Pages 6–7: lines 57–70)

3) Description of the results should be better written.

Replace “Fig. X shows” throughout the section. For instance, on page 20 you should state “we investigated/examined c-Fos expression” instead of “we observed c-Fos expression” and then “we observed” or “this revealed” instead of “Fig.3 shows”.

Response: Thank you for your valuable comment. We have revised the text in the manuscript accordingly

Changes in the text:

“To confirm peripheral sensitization by prolonged optogenetic stimulation, we examined the blue LED light irradiation hind paw withdrawal test …in the NaV1.x–ChR2 mice (Fig 2).” (Pages 19–20: lines 219–224)

“we investigated c-Fos expression in the secondary sensory neurons in the dorsal horn of the spinal cord. We observed c-Fos expression in the ipsilateral and contralateral dorsal horn of the lumber spinal cord after long term optogenetic stimulation (Fig 3).” (Page 21: lines 237–240)

“We examined the analgesic effect of 10 and 100 mg/kg DS-1971a, which is a selective NaV1.7 inhibitor [15], … in the NaV1.7-ChR2, NaV1.8-ChR2 and NaV1.9-ChR2 mice (Fig 4).” (Page 23: lines 256–261)

4) On page 21, there is no introduction about the aim of the experiment. Why studying DS-1971a effect? Why focusing on NaV1.7 and not investigating the others? Why looking at the responses of the NaV1.8 and NaV1.9 mice after DS-1971a treatment if NaV1.7 is “globally expressed”? This also goes back to better describing the expression and role of these three sodium channels in the introduction. Your description of the results (lines 241-243) is a title of a figure legend but definitively not a result description/explanation. This must be rephrased.

Additional experiments should also be done to fully support your conclusions mentioned in the discussion. It is important to examine the long-lasting effect of the NaV1.7 inhibitor using the same treatment paradigm. Given your model elicits allodynia for 24 h, does NaV1.7 inhibition prevent hypersensitivity appearance when looking at 24 h post-optogenetic stimulation or does it only block the acute phase you investigated here and simply delays sensory alterations?

In this line, NaV expressions are altered in neuropathic pain, it would be of interest to assess whether this is the case in your model 3 h, 24 h, and 48 h after light simulation during the acute phase when DS-1971a is effective, the subacute phase (to link it with the (lack of) effect of DS-1971a at that time point), and when normal sensory response is restored, respectively.

Additionally, given DS-1971a is efficient in all mouse genotypes, you would gain crucial information i/ by examining the influence of DS-1971a in hyperalgia tests in addition to allodynia and ii/ by specifically discussing the role of A-beta sensory fibres, and C vs A-delta nociceptive fibers as the three NaV channels have different expression patterns.

In this line, it would be important to test the impact of low and high TTX doses to discriminate A fibre (low dose) from A+C fibre (high dose) implications in nociceptive and sensory tests.

Also, does blocking NaV1.8 using VX-548 or LTGO-33 for instance lead to the same result in all three genotypes in different nociceptive and sensory tests?

This is crucial for the description of your model, for the understanding of the role of each sodium channel, thereby for proposing specific therapeutic targets for specific pain types.

Response: Thank you for your valuable comment. The purpose of the experiments with DS-1971a has been added. The description of the results has also been revised.

NaV1.7 is abundantly distributed in DRG and is strongly associated with pain generation, and selective NaV1.7 inhibitors have often been studied for their analgesic effects in animal models of neuropathic pain. Fortunately, we were able to use DS-1971a in this study because Daiichi Sankyo gifted us DS-1971a.

Although DS-1971a suppressed neuropathic pain in NaV1.7-ChR2 mice as expected, it was surprising that DS-1971a also suppressed neuropathic pain in NaV1.8-ChR2 and NaV1.9-ChR2 mice. As described in Discussion, since NaV1.7 is globally expressed in various types of DRG, we hypothesize that NaV1.7 is also expressed in DRG expressing NaV1.8 and NaV1.9 and that DS-1971a also suppresses these DRG.

As the analgesic effect of DS-1971a on SNL and PSL mice in previous studies was around 2-4 h [15], in the present study, we believe that DS-1971a also temporarily alleviates neuropathic pain in our model. So, we did not observe any long-term effects of DS-1971a. However, as reviewer 1 has indicated, it is possible that NaV expression is increased in our model of neuropathic pain, and initial administration of DS-1971a may provide long-term pain relief by suppressing it. To elucidate these issues, analysis of changes in NaV expression should also be performed. Therefore, we have not included these matters in this study and would like to leave them for future research.

Selective Na+ channel inhibitors against NaV1.7 and NaV1.8 exist, but there are currently no selective inhibitors against NaV1.9. As reviewer 1 has accurately pointed out, additional experiments with selective inhibitors against NaV1.8, such as VX-548, or with TTX would provide more insight, but due to cost and time constraints, we would like to limit the current report to experiments with DS-1971a. However, the fact that we were able to demonstrate the analgesic effect of DS-1971a on neuropathic pain is of great significance for future clinical practice.

Changes in the text:

“Clinical trials of selective Na+ channel inhibitors are underway for the treatment of acute postoperative pain and neuropathic pain [17]. We have examined the analgesic effect of 10 and 100 mg/kg DS-1971a, which is a selective NaV1.7 inhibitor [15], on long-term optogenetic stimulation-induced neuropathic pain in NaV1.x–ChR2 mice. DS-1971a at both 10 mg/kg and 100 mg/kg almost completely suppressed long-term optogenetic stimulation-induced neuropathic pain in the NaV1.7-ChR2, NaV1.8-ChR2 and NaV1.9-ChR2 mice (Fig 4).” (Page 23: lines 255–261)

Minor comments

1) c-Fos expression is not a pain perception marker as stated several times in the manuscript. It reflects neuronal activity. In this case linked to nociceptor stimulation yet its expression is not specific to pain. This has therefore to be modified.

Response: Thank you for your valuable comment. We have modified the text as per your suggestion.

Changes in the text:

“c-Fos, a marker for neuronal activity following noxious stimulation,” (Page 3: line 36) (Page 26: line 298)

2) Below-mentioned sentences in introduction (lines 65-67) belong to the discussion or need to be rephrased:

“Previously, Daou et al. generated NaV1.8−ChR2 mice using the NaV1.8−Cre recombinase knock-in mouse line and reported results similar to ours [10]. They also reported the occurrence of transient neuropathic pain following long-term exposure to blue light.”

Response: Thank you for your valuable comment. We have rephrased the text as per your suggestion.

Changes in the text:

“Previously, Daou et al. reported the occurrence of transient neuropathic pain following long-term exposure to blue light in NaV1.8−ChR2 mice [11].” (Pages 7–8: lines 78–80)

3) Materials & Methods, line 83. The authors mention that the experimenters are blinded to the mouse genotypes. Are they also blinded to the treatments (DS-1971a vs vehicle) used as this is also crucial?

Response: Thank you for your valuable comment. We have modified the text as per your suggestion.

Changes in the text:

“Mice in each group were randomly selected, and the experimenter was blinded to the mouse genotype and drug treatment.” (Page 9: lines 95–97)

4) Materials & Methods, lines 149-151. The below-mentioned sentence goes into the results section:

“Mice were confirmed hypersensitive by von Frey test 1 hour after a prolonged (30 min), suprathreshold (7.5 mW for NaV1.7–ChR2, 1.8 mW for NaV1.8–ChR2, and 2.25 mW for NaV1.9–ChR2) light stimulation.”

Response: Thank you for your valuable comment. However, we believe that this text should be in the Materials & Methods section. Instead, the irradiation times have been added to figure legend.

Changes in the text:

“c-Fos expression in the ipsilateral and contralateral dorsal horn of the lumber spinal cord after long term (30 min) optogenetic stimulation (7.5 mW for NaV1.7–ChR2, 1.8 mW for NaV1.8–ChR2, and 2.25 mW for NaV1.9–ChR2) are shown in (a) NaV1.7–ChR2, (b) NaV1.8–ChR2, (c) NaV1.9–ChR2 mice.” (Page 21: line 245)

5) Results. Line 208. “Continuous” should be replaced by “prolonged”.

Response: Thank you for your valuable comment. We have modified the text as per your suggestion.

Changes in the text:

“To confirm peripheral sensitization by prolonged optogenetic stimulation, we examined the blue LED light irradiation hind paw withdrawal test before and 1 hour after prolonged (30 min) optogenetic stimulation.” (Pages 19-20: lines 219-221)

6) Discussion is lacking an introductory a summary of your findings.

Response: Thank you for your valuable comment. We have added the text as per your suggestion.

Changes in the text:

“In this study, we demonstrated that a reversible neuropathic pain state was established persisting for a minimum of 24 hours when light-responsive pain mice (NaV1.x-ChR2 mice) was irradiated with light of an intensity that consistently elicited pain. Furthermore, the mice also showed hypersensitivity to light irradiation and mechanical stimulation. The c-Fos expression increased in the dorsal horn of the spinal cord on the light irradiated side. DS-1971a, a selective NaV1.7 inhibitor, was effective in attenuating neuropathic pain in all the light-responsive pain mice.” (Page 24: lines 273–279)

7) Discussion (lines 276-279). This belongs to the result section.

Response: Thank you for your valuable comment. We have added and revised the texts as per your suggestion.

Changes in the text:

“DS-1971a at both 10 mg/kg and 100 mg/kg almost completely suppressed long-term optogenetic stimulation-induced neuropathic pain in the NaV1.7-ChR2, NaV1.8-ChR2 and NaV1.9-ChR2 mice (Fig 4).” (Page 23: lines 258–261)

“In this study, the selective NaV1.7 inhibitor DS-1971a for long-term optogenetic stimulation-induced neuropathic pain completely suppressed pain in the NaV1.x-ChR2 mice.” (Pages 26–27: lines 301–303)

8) Discussion (line 286). Please give references.

Response: Thank you for your valuable comment. We have added a reference.

Changes in the text:

“…from these animal models [19].” (Page 27: line 313)

Reviewer: 2

1) Overall, this manuscript is well thought out and does give enough proof to sustain the claims.

Personally, I would just edit some parts of the text, both in sentence construction as well as in some

---

## [Decision Letter · Decision Letter 1]

16 Mar 2025

PONE-D-24-58742R1Reversible neuropathic pain model created by long-term optogenetic nociceptor stimulation using light-responsive pain micePLOS ONE

Dear Dr. Maruta,

Thank you for submitting your manuscript to PLOS ONE. After careful consideration, we feel that it has merit but does not fully meet PLOS ONE’s publication criteria as it currently stands. Therefore, we invite you to submit a revised version of the manuscript that addresses the points raised during the review process.

I would kindly suggest that you try to satisfy carefully to reviewer comments before submitting your response. 

We look forward to receiving your revised manuscript.

Kind regards,

Julian Cheron

Academic Editor

PLOS ONE

Reviewers' comments:

Reviewer's Responses to Questions

**Comments to the Author**

1. If the authors have adequately addressed your comments raised in a previous round of review and you feel that this manuscript is now acceptable for publication, you may indicate that here to bypass the “Comments to the Author” section, enter your conflict of interest statement in the “Confidential to Editor” section, and submit your "Accept" recommendation.

Reviewer #1: (No Response)

2. Is the manuscript technically sound, and do the data support the conclusions?

Reviewer #1: Partly

3. Has the statistical analysis been performed appropriately and rigorously? 

Reviewer #1: Yes

4. Have the authors made all data underlying the findings in their manuscript fully available?

Reviewer #1: Yes

5. Is the manuscript presented in an intelligible fashion and written in standard English?

Reviewer #1: Yes

6. Review Comments to the Author

Reviewer #1: (No Response)

7. PLOS authors have the option to publish the peer review history of their article (what does this mean? ). If published, this will include your full peer review and any attached files.

**Do you want your identity to be public for this peer review?** For information about this choice, including consent withdrawal, please see our Privacy Policy .

Reviewer #1: No

---

## [Author Response · Author response to Decision Letter 2]

4 Apr 2025

Dear Dr. Julian Cheron,

Thank you for giving me the opportunity to submit a revised draft of the manuscript titled “Reversible neuropathic pain model created by long-term optogenetic nociceptor stimulation using light-responsive pain mice.” to PLoS ONE (Manuscript ID: PONE-D-24-58742).

We greatly appreciate the time and effort that you and the reviewers have dedicated to providing your valuable feedback on our manuscript. We are grateful to the reviewers for their insightful comments on this paper.

We have extensively revised our paper to reflect most of the suggestions provided by the reviewers. We have highlighted the changes within the manuscript in red font in the marked version (Revised manuscript with Track Changes). Here is a point-by-point response to the reviewers’ comments and concerns.

Reviewer: 1

Major comments:

1) The use of “neuropathic pain” in the description of their model is confusing. Indeed, neuropathy is large family of pain types which can be acute, a chronic state resulting from neuronal lesion with sensory alterations, combined to inflammation or metabolic alterations... In the discussion, they compare the advantage of their acute model of nociception over sciatic nerve injury that leads to chronic neuropathy, a very different clinical picture. A more precise description of the proposed model in respect to clinical relevance therefore needs to be clarified to draw the right conclusions.

In this line, the authors should give examples and references about the other animal models of neuropathic pain they are comparing theirs to (introduction, line 53) to clarify latter the potential added value and limitations of their model. Also, they refer to a previous work of theirs introducing this optogenetic-based model (introduction, line 61-65) and mention that “nociceptive pain was produced”, not neuropathic pain….

Response: Thank you for your valuable comment. We have already provided examples and references about the other animal models of neuropathic pain, which we are comparing our model to, to clarify latter the potential added value of our model in the discussion section. Kindly see Discussion, Pages 27–29: lines 311–327, “A multitude of animal models of neuropathic pain…specific voltage-gated Na+ channels”.

The authors did not include any example nor references of other models of neuropathic pain in the introduction (as previously asked) to later compare the supposed advantages of their model.

Indeed, in the discussion, they did not get into the specificities of their model vs others in respect to the fact that existing models “may not accurately recapitulate the clinical manifestations of the disease”. I still do not see how their model addresses this question. I understand their model produces central sensitization (which may lead to neuropathic pain) and that the model can be of importance for studying the initial mechanisms of central sensitization, but there is no data supporting that this is a clinically relevant model of neuropathic pain, notably because neuropathic pain is a chronic state (for instance in the Chung model which is the only one they discuss) while sensory alterations are reversible after 24 hours in their model of acute neuropathy.

Our light-responsive pain mice (NaV1.x−ChR2) induce nociceptive pain on exposure to specific wavelengths, similar to the light-responsive pain mice (NaV1.8−ChR2) developed by Daou et al. Moreover, repeated nociceptive pain can lead to neuropathic pain due to central sensitization [16]. Daou et al. and we have also demonstrated that prolonged light exposure can cause neuropathic pain. We have already mentioned these points in the discussion section (Page 26: lines 294-301, “Daou et al. reported that…thereby leading to central sensitization [16]”).

Response 2: Thank you for your valuable comment. Conventional animal models of neuropathic pain in which peripheral nerves are physically damaged may result in motor neuropathy and sensory neuropathy other than pain. Optogenetics-based neuropathic pain models are superior to conventional animal models in that they can precisely target pain nerves. Furthermore, by targeting only the pain nerves, they may more faithfully reproduce the clinical manifestations of neuropathic pain. The text has been added in light of the above.

Changes in the text:

“The most widely employed models of neuropathic pain attributed to peripheral neuropathy in rodents are physical injury models of peripheral nerves, including the spared nerve injury (SNI), chronic constriction injury (CCI), partial sciatic nerve ligation (PSL), partial sciatic nerve ligation (PSL), and spinal nerve ligation (SNL) (Chung model) models [3]. Although these models produce allodynia, they can cause motor and/or non-pain sensory neuropathies.” (Page 5: lines 52–57)

“Such optogenetics-based neuropathic pain models are superior to conventional animal models in that they can target only the sensory nerves responsible for pain.” (Page 8: lines 85–87)

“Furthermore, although pain persisted for up to 24–36 hours in this study, more chronic neuropathic pain may occur with higher light intensity or longer exposure duration, which may contribute to the understanding of chronic pain.” (Page 30: lines 342–344)

4) On page 21, there is no introduction about the aim of the experiment. Why studying DS-1971a effect? Why focusing on NaV1.7 and not investigating the others? Why looking at the responses of the NaV1.8 and NaV1.9 mice after DS-1971a treatment if NaV1.7 is “globally expressed”? This also goes back to better describing the expression and role of these three sodium channels in the introduction. Your description of the results (lines 241-243) is a title of a figure legend but definitively not a result description/explanation. This must be rephrased.

Additional experiments should also be done to fully support your conclusions mentioned in the discussion. It is important to examine the long-lasting effect of the NaV1.7 inhibitor using the same treatment paradigm. Given your model elicits allodynia for 24 h, does NaV1.7 inhibition prevent hypersensitivity appearance when looking at 24 h post-optogenetic stimulation or does it only block the acute phase you investigated here and simply delays sensory alterations?

In this line, NaV expressions are altered in neuropathic pain, it would be of interest to assess whether this is the case in your model 3 h, 24 h, and 48 h after light simulation during the acute phase when DS-1971a is effective, the subacute phase (to link it with the (lack of) effect of DS-1971a at that time point), and when normal sensory response is restored, respectively.

Additionally, given DS-1971a is efficient in all mouse genotypes, you would gain crucial information i/ by examining the influence of DS-1971a in hyperalgia tests in addition to allodynia and ii/ by specifically discussing the role of A-beta sensory fibres, and C vs A-delta nociceptive fibers as the three NaV channels have different expression patterns.

In this line, it would be important to test the impact of low and high TTX doses to discriminate A fibre (low dose) from A+C fibre (high dose) implications in nociceptive and sensory tests.

Also, does blocking NaV1.8 using VX-548 or LTGO-33 for instance lead to the same result in all three genotypes in different nociceptive and sensory tests?

This is crucial for the description of your model, for the understanding of the role of each sodium channel, thereby for proposing specific therapeutic targets for specific pain types.

Response: Thank you for your valuable comment. The purpose of the experiments with DS-1971a has been added. The description of the results has also been revised.

NaV1.7 is abundantly distributed in DRG and is strongly associated with pain generation, and selective NaV1.7 inhibitors have often been studied for their analgesic effects in animal models of neuropathic pain. Fortunately, we were able to use DS-1971a in this study because Daiichi Sankyo gifted us DS-1971a.

Although DS-1971a suppressed neuropathic pain in NaV1.7-ChR2 mice as expected, it was surprising that DS-1971a also suppressed neuropathic pain in NaV1.8-ChR2 and NaV1.9-ChR2 mice. As described in Discussion, since NaV1.7 is globally expressed in various types of DRG, we hypothesize that NaV1.7 is also expressed in DRG expressing NaV1.8 and NaV1.9 and that DS-1971a also suppresses these DRG.

As the analgesic effect of DS-1971a on SNL and PSL mice in previous studies was around 2-4 h [15], in the present study, we believe that DS-1971a also temporarily alleviates neuropathic pain in our model. So, we did not observe any long-term effects of DS-1971a.

They authors did not observe or they did not examine? I still think it is an important experiment to see whether blocking the initial phase of nociception is sufficient to prevent central sensitization.

However, as reviewer 1 has indicated, it is possible that NaV expression is increased in our model of neuropathic pain, and initial administration of DS-1971a may provide long-term pain relief by suppressing it. To elucidate these issues, analysis of changes in NaV expression should also be performed. Therefore, we have not included these matters in this study and would like to leave them for future research.

Again DS-1971a long-term effect should be studied and depending on the results examining NaV expression changes may or may not be of importance.

Selective Na+ channel inhibitors against NaV1.7 and NaV1.8 exist, but there are currently no selective inhibitors against NaV1.9. As reviewer 1 has accurately pointed out, additional experiments with selective inhibitors against NaV1.8, such as VX-548, or with TTX would provide more insight, but due to cost and time constraints, we would like to limit the current report to experiments with DS-1971a. However, the fact that we were able to demonstrate the analgesic effect of DS-1971a on neuropathic pain is of great significance for future clinical practice.

I would have agreed if the authors had focused on NaV1.7 only but they examined the importance of several channels that could have different impact on the establishment of central sensitization. In the end they do not examine deep enough the importance of NaV1.7 nor do they investigate the importance of at least NaV1.8 (as NaV1.9 has no selective inhibitor).

As they point out DS-1971a could be of importance for clinical practice, but they cite a reference supporting this already, thus mellowing the importance of their results, and we circle back to a better understanding of their model in respect to clinical picture.

Experiments 1 and 3 are mandatory and experiment 2 remains optional depending on the result of experiment 1.

Changes in the text:

“Clinical trials of selective Na+ channel inhibitors are underway for the treatment of acute postoperative pain and neuropathic pain [17]. We have examined the analgesic effect of 10 and 100 mg/kg DS-1971a, which is a selective NaV1.7 inhibitor [15], on long-term optogenetic stimulation-induced neuropathic pain in NaV1.x–ChR2 mice. DS-1971a at both 10 mg/kg and 100 mg/kg almost completely suppressed long-term optogenetic stimulation-induced neuropathic pain in the NaV1.7-ChR2, NaV1.8-ChR2 and NaV1.9-ChR2 mice (Fig 4).” (Page 23: lines 255–261)

Response 2: Thank you for your valuable comment. As neuropathic pain in light-responsive pain mice is reversible, over time it becomes impossible to determine whether the analgesia is due to DS-1971a or to a natural process. Furthermore, the effects of DS-1971a last for 2–4 hours. Therefore, I have only observed the effects of DS-1971a for a short time immediately after the onset of neuropathic pain in this study. And unfortunately, I do not currently have temporary possession of the NaV1.x-ChR2 mice due to a change of laboratory in April this year. It is also expected that it will take some time to set up the new laboratory. Therefore, while we fully appreciate that the additional experiments recommended by reviewer would make our study more meaningful, additional experiments could not be performed. Thus, I hope that additional experiments on the long-term effects of DS-1971a and selective NaV1.8 inhibitors could be left for future research. These have been added to the Discussion section as limitations to this study.

Changes in the text:

“In the present study, we only observed the short-term analgesic effects of DS-1971a; however, successful initial treatment of neuropathic pain may lead to long-term analgesic effects. Furthermore, to understand the role of NaV1.8 in neuropathic pain, it is important to clarify how selective NaV1.8 inhibitors (although no selective NaV1.9 inhibitors currently exist) act in the respective NaV1.x-ChR2 mice. These limitations of the present study and issues should be addressed in future studies.” (Page 28: lines 319–324)

Minor comments

4) Materials & Methods, lines 149-151. The below-mentioned sentence goes into the results section:

“Mice were confirmed hypersensitive by von Frey test 1 hour after a prolonged (30 min), suprathreshold (7.5 mW for NaV1.7–ChR2, 1.8 mW for NaV1.8–ChR2, and 2.25 mW for NaV1.9–ChR2) light stimulation.”

Response: Thank you for your valuable comment. However, we believe that this text should be in the Materials & Methods section. Instead, the irradiation times have been added to figure legend.

Changes in the text:

“c-Fos expression in the ipsilateral and contralateral dorsal horn of the lumber spinal cord after long term (30 min) optogenetic stimulation (7.5 mW for NaV1.7–ChR2, 1.8 mW for NaV1.8–ChR2, and 2.25 mW for NaV1.9–ChR2) are shown in (a) NaV1.7–ChR2, (b) NaV1.8–ChR2, (c) NaV1.9–ChR2 mice.” (Page 21: line 245)

This is still not a description of the immunofluorescence protocol, but as they included the details in the figure legend, they can simplify by stating that mice that showed hypersensivity using the von Frey test were then fixed for immunofluorescence analysis.

Response 2: Thank you for your valuable comment. We have added the text in the Result section as per your suggestion.

Changes in the text:

“Mice that demonstrated hypersensitivity using the von Frey test were fixed for immunofluorescence analysis.” (Page 22: lines 245–247)

---

## [Editor Report · Decision Letter 2]

11 Apr 2025

Reversible neuropathic pain model created by long-term optogenetic nociceptor stimulation using light-responsive pain mice

PONE-D-24-58742R2

Dear Dr. Maruta

We’re pleased to inform you that your manuscript has been judged scientifically suitable for publication and will be formally accepted for publication once it meets all outstanding technical requirements.

Kind regards,

Julian Cheron

Academic Editor

PLOS ONE

---

## [Editor Report · Acceptance letter]

PONE-D-24-58742R2

PLOS ONE

Dear Dr. Maruta,

I'm pleased to inform you that your manuscript has been deemed suitable for publication in PLOS ONE. Congratulations! Your manuscript is now being handed over to our production team.

Kind regards,

on behalf of

Dr. Julian Cheron

Academic Editor

PLOS ONE